# The Matter of Captchas: An Analysis of a Brittle Security Feature on the Modern Web

## ABSTRACT

The web ecosystem is a fast-paced environment. In this dynamic landscape, new security features are offered one after another to enhance the security and robustness of web applications and the operations they handle. This paper focuses on a fragile but still in-use security feature, text-based CAPTCHAs, that had been wildly used by web applications in the past to protect against automated attacks such as credential stuffing and automated account hijacking. The paper first investigates what it takes to develop automated scanners that can solve previously unseen text-based CAPTCHAs. To this end, we evaluated the possibility of developing and integrating a pre-trained CAPTCHA solver in the automated web scanning process without using a significantly large training dataset. We also performed an analysis of the impact of such autonomous scanners on CAPTCHA-enabled websites. Our analysis showed that using solvable text-based CAPTCHAs on login, contact, and comment pages of websites is not uncommon. In particular, we identified more than 3,000 text-based CAPTCHA websites in critical sectors such as finance, government, and health, involving hundreds of thousands of users. We showed that a web scanner with a pre-trained solver could solve more than 20% of previously unseen CAPTCHAs in just one single attempt. This result is worrisome considering the substantial potential to autonomously run the operation across thousands of websites on a daily basis with minimal training. Furthermore, the finding suggests that the integration of autonomous scanning with pre-training and local optimization of models can significantly increase adversaries' asymmetric power to launch their attacks cheaper and faster.

## CCS CONCEPTS

• **Security and privacy → Web application security**; • **Information systems → Web crawling**.

## KEYWORDS

Web Security, Automated Attacks, Web Bots, Captcha

**ACM Reference Format:**
Anonymous Author(s). 2023. The Matter of Captchas: An Analysis of a Brittle Security Feature on the Modern Web . In *Proceedings of ACM Web Conference 2024 (WWW '24)*. ACM, New York, NY, USA, 13 pages. https://doi.org/XXXXXXX.XXXXXXX

## 1 INTRODUCTION

As the web continues to evolve, new features and mechanisms are introduced to improve the security and trustworthiness of web applications. In this context, deprecated security features refer to technologies or practices that have become outdated and are no longer recommended due to known vulnerabilities or possible evasions. Appcache [35] and NPAPI Plugin support [69] are just a few examples of these deprecated features that were replaced by more robust features because of their impacts on the security posture of web applications and their users. Among many other security features, text-based CAPTCHAs (Completely Automated Public Turing test to tell Computers and Humans Apart) have been also considered deprecated as a sole or primary security measure to prevent automated bots and offensive web scanners. While there has not been any formal announcement declaring text-based CAPTCHAs as officially deprecated, there has been ongoing discussion in the websecurity and web development communities [38, 60] about the limitations of text-based CAPTCHAs. Over the last few years, several new technologies (e.g., re-CAPTCHA v3 [13], behavioral analytics [9, 12], 2FA [46]) have been developed to protect web applications from offensive non-human web traffic.

There is no lack of evidence that text-based CAPTCHA has failed to achieve its guarantees [15, 37, 45], and prior work [52, 84] have proposed multiple proof of concepts to bypass this security feature. In this paper, we aim to answer how this brittle security feature can be impacted in a real-world setting by modern offensive scanners that are designed to run autonomously at scale. In particular, we will answer how much technical effort or training data is needed to build an integrated cracking method in modern web scanners to solve arbitrary text-based CAPTCHAs found in target websites. Answering these questions is important because: (1) the attack assumptions seem realistic and in line with cost-sensitive adversaries' goals – which is to build generalizable offensive tools that can be applicable to various cases with minimal effort, (2) bypassing CAPTCHAs, as a security mechanism, can lead to consequential issues — a generalizable CAPTCHA solver integrated with web scanning can facilitate automated credential stuffing and spamming at scale. Lastly, answering these questions is critical, especially with the rising threat of abusing pre-trained models in the wild (i.e., ChatGPT4 [55]).

To answer these questions, we created an analysis pipeline by first generating a large catalog of CAPTCHA challenges, including more than 15,000 images from 20 different CAPTCHA schemes. We analyzed 14 open-source CAPTCHA solvers and incorporated four well-known CAPTCHA cracking methods in our final pipeline. We also identified over 3,000 websites with solvable CAPTCHAs in their offered services. In the following, we highlight some of the major results of this paper.

**Deep learning-based object detection methods open new opportunities in this adversarial space.** While most of the solvers performed well on their own trained CAPTCHA schemes,

they often failed to handle unfamiliar, out-of-distribution, yet simple CAPTCHAs. On the other hand, our analysis showed that with a proper dataset that is not necessarily large and a proper learning algorithm, it is still possible to build pre-trained models that can solve CAPTCHAs even from unseen schemes. In particular, we observed that object detection models [21] can achieve this goal because their performance does not depend on the underlying image segmentation – the pre-processing step that has been the choke point of almost all the other solvers – to extract letters. Our analysis shows that the pre-trained model can successfully crack 13 different schemes that were not seen in the training phase. The results show that a pre-trained model can solve CAPTCHAs with a success rate of 80% where the average number of guesses in eight CAPTCHA schemes is only 2.1 attempts (See Table 2).

**The integration of pre-trained solvers into offensive scanning can lead to consequential security problems.** We performed a large-scale analysis to evaluate the effectiveness of the integrated solver in the web scanning process. To this end, we built a dataset of top-rank domains [24] plus 30K login URLs collected from prior studies [14, 66] and identified over 3,000 websites with potentially solvable CAPTCHAs on the login, contact, or comment page. The pre-trained model was able to successfully solve nearly 20% of the challenges in just one single attempt. **Due to the nature of the task, which can be launched autonomously, 20% success rate is significant when those autonomous scanners can hit thousands of websites each day to perform credential stuffing or send malicious emails and spams on target applications.** We observed that over 30% of the vulnerable CAPTCHAs belonged to websites in critical sectors such as government, finance, and health with several thousands of monthly users (See Section 5.3). We followed several ethics measures (see Section 5.1) and responsibly disclosed the issues to the website publishers five months before the submission.

While the issues discussed in this paper may not lead to severe security problems in web applications that utilize state-of-art defense services, they are a significant risk to web applications that are using them. This security feature does not provide the security guarantees it used to offer which in fact makes it even more attractive for adversaries for for automated credential stuffing and other forms of automated attacks on websites that offer important services to users and handle vital data. In addition to the problem of using a deprecated solution, our analysis shows the risk of using pre-train models in the adversarial landscape where pre-trained models can be distributed among adversaries and optimized with new data for better coverage and cracking rate. This can significantly increase adversaries' asymmetric power to launch their attacks on web applications in a cheaper, faster, and more consequential way. Our hope is that this work serves to raise awareness about the importance of removing these deprecated features at the web scale and minimizing the impacts of such unsavory practices on critical web applications.

**Contributions.** The contributions are summarized as follows:

- We build a catalog of CAPTCHA images including more than 15,000 samples from 20 different CAPTCHA schemes. We use this dataset as a reference point to evaluate the effectiveness of automated scanners.

- We develop a pre-trained solving model using object detection deep learning and show the model was able to solve 80% of the CAPTCHA challenges from 13 unseen schemes with minimal training.

- We analyze top-ranked websites from The Chrome User Experience Report (CrUX) [24] to gain insights into the prevalence of solvable CAPTCHAs in the wild.

- The pre-trained models, the augmented web crawler, and references to any used dataset are available for reviewers at https://github.com/scannerpaper/artifacts.

## 2 BACKGROUND AND MOTIVATION

### 2.1 Threat Model

In this paper, we make the following assumptions on attackers' goals and capabilities: **(1) adversaries behind offensive scanners are cost-sensitive**– they aim to mainly target websites that require less effort to evade bot prevention methods, **(2) cost-sensitive attackers mainly rely on available tools to develop their offensive tools** – the modules used to run the attacks relies on optimizing publicly accessible methods, **(3) scanning can be done via proxies** – the traffic can be generated from legitimate cloud infrastructure or compromised web hosting servers to reduce the effectiveness of common network-based reputation analysis techniques [1, 3, 4, 17, 18], **(4) scanners simulating user interaction:** malicious code can interact with the target web application to reduce the chance of being flagged as automated bot by invoking some of the dynamic behavior of the target application and exercise new program paths. Hence, our threat model in this paper will consider these and other similar risks and the solutions that will be developed should be robust against these malicious activities.

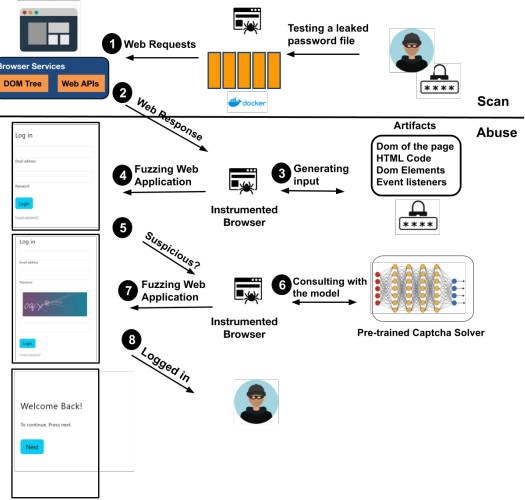

**Figure 1: A motivating example. Automated Web Scanner for credential stuffing.**

## 2.2 Credential Stuffing: A Motivating Example

Credential stuffing is the process of injecting breached credentials into websites' login forms with the goal of gaining unauthorized access to target accounts. Credential stuffing [74] has been an effective attack vector in the wild to gain unauthorized access to an account. The central assumption in credential stuffing is that password reuse is a common practice among technically less sophisticated users and even if strong passwords are enforced, trying leaked credentials across other services can potentially lead to a compromise. One challenge to running successful credential stuffing on web applications is to be able to successfully solve CAPTCHA challenges when asked. Having access to a list of leaked usernames and passwords as well as login pages of target web applications, an adversary can run a distributed cracking operation, while satisfying the checks illustrated in Figure 1. We have created a demo[1] that shows how an autonomous scanner with an embedded CAPTCHA cracking module can impact the security posture of websites using this fragile security feature. The demo is posted anonymously.

## 2.3 The Focus of This Paper

The focus of this paper is twofold: First, we aim to empirically evaluate what it would cost to make web scanners capable of solving unseen text-based CAPTCHAs – to extend the coverage of automated attacks (e.g., credential stuffing) at the web scale. We plan to answer how a cost-sensitive adversary can construct a CAPTCHA cracking model with minimal access to training data, incorporate the model into the scanning process, and start automated scanning at scale. Second, we also aim to answer what would be the impact of an efficient text-based CAPTCHA solver at the web scale where there do still exist critical web applications with vulnerable text-based CAPTCHAs.

## 3 METHODOLOGY

In this section, we elaborate on how we conducted our research to construct a dataset and build a catalog of available algorithms to construct the pre-trained models and test their effectiveness on unseen and out-of-distribution CAPTCHAs.

## 3.1 Building a Catalog of CAPTCHA Challenges

As a first step, we define procedures for generating a catalog of CAPTCHA challenges for benchmarking and evaluation. The available datasets are presented in two segments: (1) a custom dataset that covers various types of CAPTCHAs with different complexity levels, and (2) the dataset from prior work [52, 78, 84]. We generated a list of CAPTCHA challenges by incorporating CAPTCHAgen library [53] an open-source configurable CAPTCHA generator. In the following, we describe some of the techniques we used to generate the CAPTCHA challenges for training and evaluation.

**Creating Background:** Many CAPTCHA generators use a monochromatic black or white background, which can be exploited by CAPTCHA crackers to easily identify the foreground pixels. In this dataset, every background image comprises a range of different colors. However, if the pixel values of the background vary too wildly, the final CAPTCHA will be extremely difficult to read. Thus, we need to

make sure the colors only vary smoothly across the image. Specifically, for each background, we randomly select 2 colors, each of which is defined as a vector of RGB values. We let the color be constant vertically, set the 2 randomized colors to be the colors at the left and right end of the image, and set the colors at other positions by linear interpolation between the 2 sides.

**Length Complexity:** For each image, we generate an alphanumeric string with a random length of between 5 to 9 characters. For each character in the string, we create an image of that single character with random font, color, and size. Then, we apply a random rotation and a random perspective transform to the character image before pasting the character in the previously created background. The positions to paste the characters are decided as follows. Starting from the left side of the image, for each character, we randomize an offset value, then place the character that amount of pixels to the right of the previous character. This allows us to have various spacing between characters. The offset can also have a negative value, in which case we will make the character overlap with the previous one. Vertically, each character is placed close to the middle of the image, with a small random offset.

**Adding Background Noise:** Having access to a diverse set of images with texts, we make these CAPTCHA images more difficult to solve by adding some random noise. For each image, we randomly draw some curves and dots with random colors. The number of curves and dots as well as their thickness is carefully tuned so that they make it significantly more difficult to segment the text pixels from the background pixels while not corrupting the image.

**The dataset.** We have generated CAPTCHA samples with different security features to use in our experiments. An important feature of the generated CAPTCHA Catalog is that it includes the position of the characters as the label beside the actual CAPTCHA text. The schemes are extracted from real-world websites, both from the surface web and dark web, and manually labeled for further research. Each scheme contains different combinations of anti-segmentation techniques. Each scheme includes at least 500 images. The whole dataset consists of more than 15,000 CAPTCHA challenges across 20 different CAPTCHA schemes including those we generated ourselves for this paper as well as the ones used in recent prior work.

In addition to this custom dataset discussed here, we contacted the authors of prior work [78, 84] and used their dataset in the evaluation. Table 4 in Appendix A.2 shows the full list of all CAPTCHA schemes used in this paper along with their security features.

## 3.2 Building a Catalog of CAPTCHA Solvers

Table 1 shows the CAPTCHA solvers we used to start the experiments. The repositories are selected based on their popularity and the last update. Solving a CAPTCHA challenge consists of three interconnected steps: *Preprocessing*, *Segmentation*, and *Recognition*. Selecting one particular implementation in one step can significantly influence the effectiveness of subsequent steps. In the following, we briefly describe each step.

**Preprocessing.** In the preprocessing phase, the solver will remove any kind of noise from the image while keeping the main characters intact. Choosing "the best method" for preprocessing significantly depends on the target CAPTCHA scheme and the security features used in the images (e.g., noise level and color range).

---

[1]https://youtu.be/jzFUp5m2G6k

**Table 1: Details of Publicly Available CAPTCHA Solvers. Each solver is aimed to work with their CAPTCHA scheme, which defines the visual style and anti-segmentation techniques of the CAPTCHAs. Nine repositories contained the training dataset or the code to generate the dataset, and only six repositories shared the actual model. The security features column shows whether or not the used scheme includes Overlapping characters (O), Line Noise (L), Curve Noise (C), or Background Noise (N). Repositories marked with ☑ provide the code to generate the dataset.**

| CAPTCHA Solver | Last update | Forks | Stars | Segmentation Technique | Recognition Model | CAPTCHA Scheme | Security Features | Dataset Available | Model Available |
|---|---|---|---|---|---|---|---|---|---|
| zakizhou/CAPTCHA [82] | Nov, 2017 | 23 | 61 | - | CNN | Python CAPTCHA [6] | O-C-N | ☑ | ✗ |
| nladuo/CAPTCHA-break [51] | Nov, 2018 | 227 | 685 | Static | OCR, CNN | Custom | O-C-N | ✓ | ✗ |
| ypwhs/CAPTCHA_break [79] | Jan, 2020 | 666 | 2.4k | - | CNN, GRU | Python CAPTCHA | O-C-N | ✗ | ✓ |
| zhengwh/CAPTCHA-svm [86] | Aug, 2017 | 93 | 200 | Static | SVM | Custom | N | ✗ | ✓ |
| ptigas/simple-CAPTCHA-solver [57] | Nov, 2017 | 133 | 483 | Static | Static Matching | - | L | ✗ | ✗ |
| 0b01/SimGAN-CAPTCHA [54] | Jul. 2017 | 85 | 418 | - | CNN | Generated by GAN | O-L | ✗ | ✗ |
| JackonYang/CAPTCHA-tensorflow [33] | May, 2022 | 243 | 835 | - | CNN | Python CAPTCHA | O-C-N | ☑ | ✗ |
| PatrickLib/CAPTCHA_recognize [59] | Jun, 2017 | 167 | 502 | - | CNN | Python CAPTCHA | O-C-N | ☑ | ✗ |
| chxj1992/CAPTCHA_cracker [10] | Oct, 2019 | 52 | 157 | Static | CNN | PHP CAPTCHA | L-N | ☑ | ✓ |
| JasonLiTW/railway-CAPTCHA-solver [34] | Jul, 2018 | 153 | 692 | - | CNN | Taiwan Railway Booking | N | ☑ | ✗ |
| lllcho/CAPTCHA-breaking [45] | Jun, 2016 | 84 | 228 | - | CNN | English and Chinese | L-N | ✓ | ✓ |
| johnnyzn/DW-GAN [37] | Jan, 2022 | 21 | 141 | Contours, Static | GAN, CNN | Python CAPTCHA | O-C-N | ☑ | ✓ |
| DrMahdiRezaei/Deep-CAPTCHA/ [15] | Mar, 2021 | 7 | 33 | - | CNN | Python CAPTCHA | O-C-N | ✗ | ✗ |
| Object Detection-Based Model [21] | - | - | - | - | YOLOX | Python CAPTCHA | O-L-C-N | ☑ | ✓ |

**Segmentation.** Segmentation is the process of decomposing a given CAPTCHA image into a set of sub-images of individual characters. This step, if performed accurately, makes the recognition step much easier for the solver by providing the ability to use a wide range of models that are trained with single-character images. However, coming up with accurately segmented characters is not a trivial task and highly depends on the CAPTCHA features.

**Recognition.** This step is responsible for generating the text output of the solver. If the solver includes a segmentation step, the input of the recognition step is the single-character images. Otherwise, the recognition is performed on the whole input CAPTCHA image.

*3.2.1 Publicly Available CAPTCHA Solvers.* In eight of the solvers discussed in Table 1, the corresponding model was not available or it was not clear how to attain or create the labeled dataset. Consequently, a part of the analysis was to train the model utilizing the available dataset, thereby assessing both the accuracy and viability for seamless integration within the scanning process. The process of performing training required handling several corner cases. Suppose that we train a model for solving four-character CAPTCHAs which only include small letters. In this case, the output of the model will be 4*26 (4 characters, 26 letters of the alphabet). Therefore, the same model cannot be used for CAPTCHA schemes with different sizes or those containing capital letters and/or digits. To partially solve the issue for training, we had to resize the input image to our predefined width and height when training the model. Furthermore, we used a comprehensive dictionary containing lowercase letters, uppercase letters, and digits when encoding the CAPTCHA text to a vector. Additionally, we faced more limitations when cracking a CAPTCHA with a size other than the one used for training which caused a fundamental issue.

*3.2.2 Prior Scientific Work.* We contacted the authors of DeepCAPTCHA [52] and DW-GAN [84] and received the source code for analysis. In DeepCAPTCHA, the preprocessing stage is done by converting the image to grayscale. The model consists of multiple Softmax layers with each layer responsible for recognizing a single character. The concatenation of the outputs produces the final CAPTCHA string. The DW-GAN project makes use of generative adversarial networks (GAN) in the preprocessing stage. Since the method performs the segmentation by finding contours, it is capable of solving CAPTCHAs with variable character lengths. The segmentation phase receives a denoised image and runs techniques such as border tracing [56] and pixel-level segmentation [67] to extract characters.

We also borrowed recent deep learning-based object detection models [80] that are known to work well in extracting semantic-level features of images. In particular, we used the YOLOX model [21], a newly proposed method that can achieve high accuracy without introducing significant computational costs. Unlike classification models which only predict the class of a single object in an image, object detection models work on images with a variable number of objects. For each object in the image, the model predicts both its location in the image, represented by a bounding box as well as its class which is the character text. Using object detection removes the requirement for having accurate segmentation which has been the main flaw of current solvers.

## 4 SOLVING CAPTCHAS WITH MINIMAL TRAINING

The main objective of this paper is to evaluate the possibility of integrating trained solvers discussed in Section 3.2 and identify opportunities that exist for an adversary to build more autonomous web scanning To this end, we pre-trained the discussed CAPTCHA models using only one of the CAPTCHA schemes as the reference dataset and trained the model using the selected scheme to simulate real-world scenarios where adversaries cannot have access to all forms of training data, and it is very likely that the web scanner is exposed to CAPTCHAs from previously unseen CAPTCHA generators (out-of-distribution data). After training the model on the reference dataset, we used the model to solve CAPTCHAs from other schemes. The pre-trained CAPTCHA solvers are integrated into the scanner by importing the required files and trained model as a Python package. In the following, we briefly provide the results of those experiments.

## 4.1 Solving CAPTCHAs via Publicly Available Solvers

As mentioned earlier, in eight CAPTCHA solvers shown in table 1, the corresponding model was not available or there was minimal explanation on how to use the code to train the model. We were able to modify most of the code base in these methods to use the generated labeled dataset for training. One of the most common modifications to enable the code was to modify the pre-processing module to parse and segment inputs with variable sizes and match them to the target CAPTCHA scheme (as mentioned earlier in Section 3.2.1). Despite all the efforts, the results of the analysis were not very promising. That is, the methods were customized to be used on specific forms of CAPTCHAs, and common hardening tricks such as background noise could highly impact their accuracy.

We also incorporated DeepCAPTCHA [52] and DW-GAN [84] models as the two AI-based solvers proposed in prior work. To evaluate DeepCAPTCHA with minimal training, we used the *Python CAPTCHA* scheme as the reference scheme to train the model. We then used the model to solve 20 CAPTCHA schemes with 1000 samples for the synthesized CAPTCHA Catalog and 500 samples for the schemes from related works. Our analysis shows that while these methods were working well on their reference datasets, used for training, their accuracy dropped dramatically when used on other datasets where the CAPTCHA scheme was unknown to the model. For instance, the trained model of DeepCAPTCHA was able to correctly solve CAPTCHA of the same scheme in the test phase with an accuracy of 98%. For the DW-GAN, we used the *Rescator 1* to train the model and used the pre-trained GAN model available in the repository for the preprocessing stage. The accuracy of the model of the same CAPTCHA scheme in the test phase was around 75%. The accuracy of both models dropped significantly when applied to unseen CAPTCHA cases. In particular, we observed that in cases where the CAPTCHA contains **overlapping characters** or incorporates **background noise**, the effectiveness of these solvers drops rapidly (see figure 2).

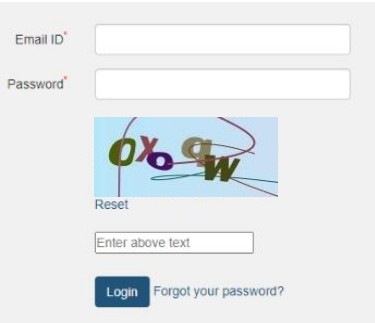

**Figure 2: A CAPTCHA in login page which fails current methods. The CAPTCHA sample contains overlapping characters and background noise.**

## 4.2 Object Detection as the Default Model

The underlying issue of the traditional segmentation techniques is that they can only process the image at the pixel level, hence they often fail in situations where characters are too close or have overlapping pixels. The common technique to mitigate this problem is to remove the noise so that it is easier to detect the characters, but since it is nontrivial to know in advance which pixels belong to characters and which ones are noise, this can lead to corrupting the object of interest and make it impossible to recognize later. In the object detection model, discussed in Section 3.2.2, each character in the image is treated as an object. Unlike classification models which only predict the class of a single object in the image, object detection models can work on images with a variable number of objects. For each object in the image, the model predicts both its location in the image, represented by a bounding box, and its class.

The training phase of the object detection model is performed using the Cat 5 scheme of our CAPTCHA Catalog. This scheme has the highest level of security features which makes the model more robust for generalized cases and provides a higher chance of solving CAPTCHAs from different schemes. It also provides the position of the characters as the label which is required for training our object detection model. We trained the model on NVIDIA RTX A5000 GPU for 10 epochs on 10K data samples in 15 minutes.

In order to evaluate our chosen object detection pre-trained model, we defined a set of experiments to run on all CAPTCHA schemes. We define a *max attempts* parameter which represents the maximum number of times the scanner is allowed to try solving the chosen CAPTCHA. Each experiment is considered successful if the solver can crack the randomly chosen CAPTCHAs before reaching the *max attempts*. Table 2 provides the results of 100 experiments when using different numbers as *max attempts*. The model was able to solve 13 CAPTCHA schemes other than the reference scheme.

**Table 2: Results of our Object detection model. Success Rate and Average Attempts of Scanner in 100 CAPTCHA solving Experiments for each scheme.**

| Max Attempts | 5 | | 20 | | 50 | |
|---|---|---|---|---|---|---|
| Scheme | Success Rate | Average Attempts | Success Rate | Average Attempts | Success Rate | Average Attempts |
| Alipay | 89.00% | 2.08 | 100.00% | 2.83 | 100.00% | 2.56 |
| Cat 1 | 100.00% | 1.62 | 100.00% | 1.44 | 100.00% | 1.56 |
| Cat 2 | 98.00% | 1.77 | 100.00% | 2.12 | 100.00% | 1.81 |
| Cat 3 | 84.00% | 2.13 | 100.00% | 3.12 | 100.00% | 3.12 |
| Cat 4 | 84.00% | 2.38 | 100.00% | 3.9 | 100.00% | 3.76 |
| Ebay | 82.00% | 2.14 | 99.00% | 2.92 | 100.00% | 3.17 |
| Live | 5.00% | 3.80 | 21.00% | 10.33 | 40.00% | 23.6 |
| Rescator 1 | 92.00% | 2.02 | 100.00% | 2.52 | 100.00% | 2.14 |
| Rescator 2 | 16.00% | 2.87 | 54.00% | 9.98 | 84.00% | 18.02 |
| Sohu | 5.00% | 2.20 | 20.00% | 10.2 | 51.00% | 22.49 |
| PythonCAPTCHA | 15.00% | 2.60 | 47.00% | 9.32 | 93.00% | 17.95 |
| Weibo | 55.00% | 2.54 | 98.00% | 6.24 | 100.00% | 5.99 |
| Yellow brick | 94.00% | 2.23 | 100.00% | 2.73 | 100.00% | 2.5 |
| **Overall** | **63.00%** | **2.33** | **79.92%** | **5.2** | **89.85%** | **8.36** |

**Missing Cases.** The analysis reveals the pre-trained model fails under specific scenarios. In this part, we discuss the root cause of false positive cases. Figure 6 in Appendix A.5 provides samples of CAPTCHAs that the model was not able to solve for different reasons. One source of false output has been aggressive background noise. Schemes *jd* and *qihu* utilize this method to defend against crackers. In other cases, unusual noise was observed in the CAPTCHA. For example, *baidu* scheme contains a specific noise pattern that is interpreted as a set of characters, resulting

in incorrect results. Furthermore, if the CAPTCHA included characters with a different style than what the model was trained on, it would also fail. Examples of such cases are the *google* and *wiki* schemes. To successfully solve CAPTCHAs with different styles, the model can be trained on that particular style of characters. However, effectively handling unusual noise requires a more advanced preprocessing stage, which is beyond the scope of this paper. Also, note that in a real-world setting, injecting extreme noise can have a significant usability effect which has been discussed extensively in prior work [8, 76].

## 5 PREVALENCE OF LEGACY CAPTCHAS ON MODERN WEB

In this section, we seek to answer how text-based CAPTCHAs are still relevant at the web scale and how the insights we gained in the previous sections can be translated into actionable tasks. To this end, we performed a large-scale crawling experiment to locate CAPTCHA-protected websites. Then, we ran an experiment using the integrated solver to check the effectiveness of the solver.

### 5.1 Ethical Considerations.

We had the following ethical considerations to conduct the experiments. First, we followed a strict minimum interaction policy with all target websites. This entailed refraining from overloading target web applications by submitting only one request to the server, even in cases where CAPTCHAs were found to be vulnerable. To solve identified CAPTCHAs on the crawled websites, we generated an offline version of the application and ran the experiment on the locally hosted website. That said, no active CAPTCHA solving or repetitive requests were generated on the target websites. Second, We contacted all the impacted websites with the deprecated security feature, notifying them about our findings several months before the submission of this paper. The detail of this process is explained in Appendix A.7. While few publishers acknowledged the issue and requested additional information on how to solve the issue, in the rest of the cases, the impacted pages with the vulnerable CAPTCHAs became unreachable or modified without any explicit acknowledgment. Third, in an effort to prevent any potential issues for website owners, we have excluded the list of vulnerable domains, along with their corresponding CAPTCHAs, from our published artifacts. This approach ensures that website owners are not impacted by our findings in any negative way.

### 5.2 Data Seeds

The starting stage of the pipeline is to access a list of target URLs to be crawled looking for text CAPTCHAs. We have separated our dataseed into two categories:

1) *Landing pages* from CrUX Top 1M dataset [24] which includes the top million most popular websites based on the Chrome User Experience Report (CrUX). The original dataset which is publicly accessible includes over 15M URLs. The dataset provides many attributes including the origin (URL) of the pages for every month [81]. We have used the data from December 2022 in our crawling pipeline. Later in this section, we discuss more details on how we have used the collected data in our analysis.

2) *Login pages* from PILU-90K dataset [66]. The dataset contains 90K URLs including index, login, and phishing pages. The authors identified the login URLs by utilizing Quantcast [58] top million websites between November 2019 to February 2020, crawled the pages of each website, and searched for password fields in the forms. The other dataseed for login pages was the dataset utilized in [14]. The authors have analyzed over a million domains from Alexa top ranking list and conducted the experiments between May to October 2019. This dataset provides a group of 20K URLs collected by doing a breadth-first search with depth two on addresses from [22].

### 5.3 Analysis Results

Despite encountering unresponsive domains in over 30% of the cases during our analysis, we were able to identify over 3,000 websites that utilize text-based CAPTCHAs. It is worth noting that the CrUX 1M dataseed includes domains that mostly point to landing pages of websites, rather than login or other forms that are more likely to include CAPTCHAs. In the following, we provide details on the nature of the marked websites regarding their category and involved users. Figure 3 illustrates examples of the usage of text-based CAPTCHAs on high-profile websites in the government and finance sectors with hundreds of thousands of users.

**Application Categories.** To gather additional insights on websites that utilize text CAPTCHAs, we leveraged the results obtained from SimilarWeb [63] to determine the category of each website. The distribution of websites across different categories is illustrated in Figure 5 in Appendix 4. We identified 25 web categories. Our analysis showed that 46% of websites that use text CAPTCHAs for security purposes belong to the categories including government, technology, finance, and education. The most critical categories, such as Law and government, Finance and Health, cover 29% of the detected CAPTCHAs.

**Involved Users.** We conducted an analysis by analyzing the number of monthly visits to websites belonging to the top four categories. The cumulative distribution function (CDF) graph in Figure 4 showcases the extensive range of monthly visits received by websites in these categories. It's important to note that the graph only displays four categories for brevity. The analysis shows that these websites are fairly high-profile in terms of observed traffic. For instance, about 50% of the websites in each category usually receive more than 100K monthly users. This analysis provides an approximate view of the potential impacts of possible abuses, shedding light on the consequences of service exploitation on involved users.

**Collection and Labeling** After identifying websites with text-based CAPTCHAs, we downloaded images with CAPTCHA attributes from the websites. After cleaning the collected data, we were able to successfully verify 1,600 unique CAPTCHAs. For this set of CAPTCHAs, we manually labeled them and assigned their complexity level based on their features. In the following, we delve into the security features employed in the collected CAPTCHAs and evaluate the performance of our proposed CAPTCHA cracker on these features. This analysis provides insights into the effectiveness of current security measures in protecting against automated attacks and sheds light on potential vulnerabilities that could be exploited by adversaries.

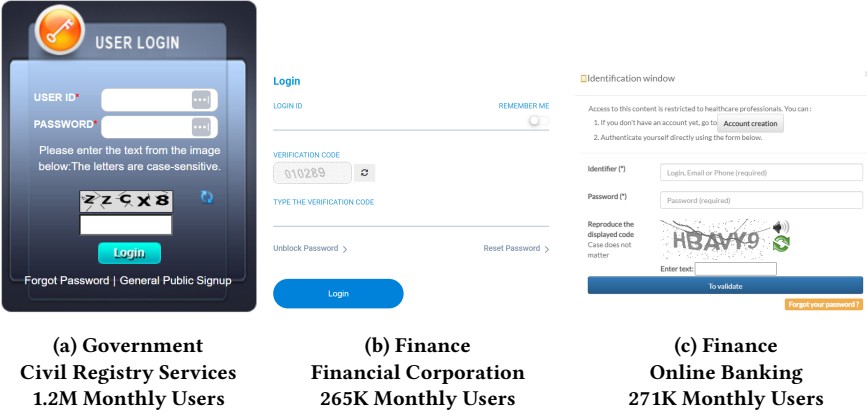

**(a) Government**
**Civil Registry Services**
**1.2M Monthly Users**

**(b) Finance**
**Financial Corporation**
**265K Monthly Users**

**(c) Finance**
**Online Banking**
**271K Monthly Users**

**Figure 3: Weak CAPTCHAs on critical web applications. Examples of using crackable CAPTCHAs on the login page of online banking and Government websites with tens of thousands of monthly visits.**

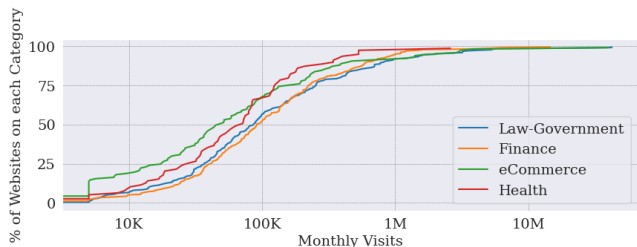

**Figure 4: CDF Diagram of Monthly Visits per Category. 50% of the websites with vulnerable CAPTCHAs have more than 100K monthly visits.**

**Level of Complexity.** We labeled the collected CAPTCHAs in seven different complexity levels. While we observed Non-English and less common CAPTCHAs (e.g., Math), over 77% of the CAPTCHAs utilize either simple colors or background noise in the generated CAPTCHAs. Table 5, in the Appendix, shows an overview of these security features and their usage rate based on the collection corpus alongside samples for each group. It is worth noting that while adding excessive noise or using different character styles for text CAPTCHAs can enhance their security, it may also negatively impact the user experience, as legitimate users also find the CAPTCHAs more difficult to solve as well.

**Solving CAPTCHAs in the Wild.** In this section, we provide a detailed analysis of the CAPTCHAs collected from various websites and the results of the attempts to crack them using the pre-trained solver, discussed in Section 4.2. To evaluate the performance with a single attempt, we used the Levenshtein distance metric [75]. Table 3 presents the distribution of distance values among different categories. The results are categorized into different buckets based on the edit distance between the true label and the predicted value. A distance of 0 indicated that the CAPTCHA was solved correctly, while a distance of at most 2 meant that the cracker had recognized the scheme but failed to identify certain characters due to the overlap of noises and characters. These cases were more likely to be cracked after multiple attempts. Upon analyzing cases

with a distance of at most 4, we found that the errors were mainly due to failure in denoising the image. Therefore, improving the preprocessing stage of the cracker can significantly impact the success of cracking these schemes. Finally, we defined the bucket with more than 4 errors, which mostly included CAPTCHAs with excessive noise, unfamiliar styles, math CAPTCHAs, and non-English CAPTCHAs. In summary, our attempts to crack CAPTCHAs collected from websites in the wild faced several challenges, primarily due to the absence of verified labels. However, our analysis using the Levenshtein distance metric provided insights into the performance of the cracker and highlighted potential areas for improvement in the preprocessing stage. The different buckets we defined based on the edit distance values allowed us to categorize the results and identify the types of CAPTCHAs that were difficult to crack.

**Table 3: Distribution of distance values using Levenshtein Distance metric. Result of the integrated CAPTCHA solver on different web categories.**

|  | Government | Finance | Health | eCommerce | Others |
|---|---|---|---|---|---|
| LD[1] = 0 | **24.81%** | **20.00%** | **18.42%** | **13.20%** | **19.13%** |
| LD ≤ 2 | **48.87%** | **42.50%** | **52.63%** | **46.78%** | **40.72%** |
| LD ≤ 4 | 18.79% | 27.5% | 26.31% | 33.96% | 29.43% |
| LD ≥ 5 | 7.51% | 10.00% | 2.63% | 11.32% | 10.72% |
| Total | 133 | 80 | 53 | 38 | 1321 |

[1] Levenshtein Distance

## 6 DISCUSSION

We here put the results of experiments into a bigger context and discuss how they can influence research on automated web attacks.
**Web Applications with Text-Based CAPTCHAs** In this paper, we identified more than 3,000 websites with text-based CAPTCHAs. At first glance, the number may seem small. However, note that our method to identify websites with text-based CAPTCHAs is a *best-effort* approach. In particular, we had to define a trade-off between launching a comprehensive experiment to identify CAPTCHA-based websites and following ethics. That is, among other factors,

many websites trigger CAPTCHAs when the remote agent manifests aggressive behavior. For ethical reasons, we avoided any interaction that could overload or make excessive requests to websites. We hope the reader acknowledges that triggering such responses requires deeper interactions with target websites, which we *intentionally* avoided to minimize potential harm. However, we showed with minimal interaction (only with the landing page), we identified this deprecated feature in various high-profile websites.

**Relevance of the Study in Today's Modern Web.** We acknowledge that contemporary security methods such as ReCAPTCHA [13] and other behavioral-based methods [9, 12] are the default mechanisms for defending against automated web traffic. However, at the global scale, text-based CAPTCHAs are still being used in websites that offer important services and maintain important user data. In particular, over 50% of the CAPTCHAs were from websites that involve over 100K monthly users (Section 5.3) including admin panels in critical sectors such as financial and healthcare. In a broader context, text-based CAPTCHAs is another example of a brittle security mechanism that is still in use in many websites – imposing virulent risks on the security and privacy of their users. Unfortunately, addressing issues of this sort on today's web has never been easy because it often requires community consensus on where and how auditing mechanisms should take place. Furthermore, defining the right incentives among entities has never been a trivial task. However, a coalition among entities such as cloud service providers that host a large group of these websites, web publishers, and developers is necessary to make awareness and maintain the security posture of web applications and their users.

**Autonomous Scanning on Known Security Problems.** There has been significant progress on the attackers' side to make large-scale web attacks more effective while evading detection. The integration of pre-trained CAPTCHA solvers in the scanning pipeline is, in fact, taking one more step in that direction. While the results of this study show the possibility of low-cost integration of pre-trained models, a more concerning trend is the integration of robust transfer learning and local optimization methods in this domain, making offensive scans even more effective over time. Our empirical analysis shows that developing more intelligent and evasive web scanners is a fairly straightforward process. With more training data and incremental improvement among adversaries, pre-trained models are likely to become a serious threat at the web scale.

## 7 RELATED WORK

In this section, we present a review of related work on the implications of deprecated web security features, web scanning as well as detection mechanisms.

**Security Analysis of Deprecated Features.** Prior work has discussed the security issues in deprecated features in software systems. Mirian et al. [47] studied the deprecation process for 2.5 years of web features in Chrome to answer how this process happens. Goethen et al. [23] proposed a security mechanism to standardize the deprecation and removal of risky web security features. Other work focused on the deprecation process on Java applications by investigating how API deprecation could lead to vulnerable code examples in platforms such as StackOverflow [62] or how API deprecation can impact client applications [87].

**Evading Anti-Scanning Defenses.** Prior research has shown several different ways that adversaries use to evade fingerprinting efforts. In [19, 20, 50, 64, 65, 70, 73], authors discuss how scanners can bypass fingerprinting techniques such as user-agent, cookies, and extensions list. Similar studies including [19, 44, 49] bring the same mindset to the operating system and hardware features such as the platform, timezone, and WebGL renderer. In [32], Invernizzi et. al. have analyzed blackhat cloaking methods used in prominent cloaking services. They use the results to propose an anti-cloaking system. In [85], authors have done an empirical study on client-side methods adapted by phishing pages in order to evade the ecosystem's detection methods. The goal of our work is to take a step further and equip the scanners with functionalities that make detection even harder through pre-trained CAPTCHA solvers.

**Attacking Challenge-based Defenses.** Challenge-based defense mechanisms have been a main line of defense against automated attacks. Tang et al. [68] present an architecture that takes the pre-processed CAPTCHA images as the input and aims to identify the number of characters using a CNN model. They collected various CAPTCHAs from the 50 most popular websites and tested each of the CAPTCHAs against the attack model they developed. In [29] and [28], Gao et al. present other dynamic approaches for the segmentation stage. The paper includes extracting components using a Log-Gabor filter followed by partition and recognition. Log-Gabor filters are applied to the images directly and extract the characters in CAPTCHA images in all four directions. Other techniques [5, 40] incorporate a two-stage pipeline for the segmentation and recognition phase and propose a single step that does both objectives. Zhang et al. [84] incorporated a generative adversarial network (GAN) model to solve a dataset of CAPTCHA images they collected after running a Tor-based crawling experiment.

**Difference from Existing Work.** The major difference between this and the prior work is that we put the problem of CAPTCHA solving into the context of adversarial web scanning. Instead of proposing an offline method to solve CAPTCHAs, this study aims to answer a different question: how an adversary can benefit from advances in learning domains and develop models with minimal training and data dependency. Answering this question is critical to empirically measure the effectiveness of modern adversarial scanners and inform threat modeling in web attacks.

## 8 CONCLUSIONS

In this paper, we evaluated the possibility of incorporating available methods in solving a deprecated security feature – text-based CAPTCHAs. We aimed to answer what would it take to build an effective and generalizable method when access to training data is minimal. We observed that object detection methods can be weaponized by adversaries for adversarial purposes as they achieve a success rate of 80% across several CAPTCHA schemes with minimal training. We identified several websites with thousands of users that can be impacted. That is, with a pre-trained model, we were able to solve 20% of those CAPTCHAs in a single attempt with no human intervention in the loop, showing the potential usage of such methods in autonomous offensive scanners in the wild.

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

# A  APPENDIX

## A.1  Current Defense Layers

We focus on three main layers of defense against automated web traffic. We explain each layer and discuss their effectiveness and potential weaknesses.

**The Role of Traffic Attribution.** An emerging defense mechanism to identify web scanners is to incorporate certain aspects of incoming web traffic as signals to detect automated scanning operations. Features such as sudden spikes in pageviews, high bounce rate (i.e., visiting a single page without clicking anything on the page), and low session duration through pages have been used in current solutions [9, 11]. Although integrating these techniques in the defense side is useful in detecting more *aggressive* scanners and bots, they cannot reveal much detail about more evasive scanners that operate more in a more targeted and non-aggressive fashion. In fact, in the most complex and consequential situations, evasive scanners can bypass almost all of these defense mechanisms by injecting artificial delays between consecutive requests, as mentioned in the threat model, or by artificially increasing the session duration by automatically triggering specific events periodically. Furthermore, we expect that the diversity of new forms of web attacks will continue to increase since attacks are becoming continuously more mature in imitating human interaction, making it increasingly more difficult to analyze the intent of remote agents.

**The Role of Fingerprinting.** Conventional fingerprinting techniques [17, 42, 50, 83] gather specific details about connecting devices such as user agents, hardware details, geolocation, and generate a unique id for that device. Fingerprints can be used for several purposes such as detecting previously seen adversarial agents attempting to visit the target web application. If an agent with automated activity is detected, the corresponding fingerprint will be

used in subsequent checks. While fingerprinting mechanisms have been useful in tracking real users, they offer limited capabilities to identify modern evasive attacks. In particular, it is not very difficult to generate malicious code that can report any arbitrary identity or establish connections from different parts of the web while launching attacks in a stealthy way [31, 36, 71] – as mentioned in the threat model. For instance, it is very common to incorporate various network proxies and use less suspicious traffic sources, use full-fledged browsers instead of using curl or headless browsers, report arbitrary information while interacting with the web application, making the current defense mechanisms less effective in detecting more evasive cases.

**The Role of CAPTCHAs.** Solving a CAPTCHA challenge has been one of the main layers of defense against automated scanners and offensive bots. Suspicious agents are required to solve a CAPTCHA challenge by extracting the content of a distorted image, object, or audio file to access the desired service. The underlying assumption is that human users can extract letters or identify objects easier than scanners and non-human agents. Image-based CAPTCHA, reCAPTCHA v2 [25] and hCAPTCHA [30] are the main techniques widely used in practice. One of the main drawbacks of CAPTCHA challenges, in general, is that the process is significantly dependent on real users. In fact, dedicating the main defense responsibility to legitimate users is not very in line with important security design principles such as psychological acceptability and the economy of mechanism. Unfortunately, many users, including non-English speaking users and special groups, have reported difficulties in solving specific forms of CAPTCHA challenges [26, 48, 61, 77].

## A.2 CAPTCHA Schemes

**Table 4: All CAPTCHA schemes used in our evaluations.**

| Scheme | Sample | Security Features |
|---|---|---|
| **Cat 1** | | - Random color for characters |
| **Cat 2** | | - Random color for characters
- Overlapping |
| **Cat 3** | | - Random color for characters
- Random color arc lines |
| **Cat 4** | | - Random color for characters
- Overlapping
- Random color arc lines |
| **Cat 5** | | - Random color for characters
- Background dots and arc lines
- Different fonts lines
- Different number of characters |
| **Python CAPTCHA** [6] | | - Background dots and horizontal lines
- Overlapping |
| **Yellow Brick** [37] | | - Background dots and horizontal lines |
| **rescator_1** [37] | | - Random color for characters
- Random color horizontal lines |
| **rescator_2** [37] | | - Horizontal lines |
| **alipay** [78] | | - Styling
- Overlapping |
| **baidu** [78] | | - Overlapping
- Arc lines |
| **ebay** [78] | | - Overlapping |
| **google** [78] | | - Styling
- Overlapping |
| **jd** [78] | | - Overlapping
- Background pattern |
| **live** [78] | | - Styling
- Character deformation |
| **qihu** [78] | | - Styling
- Overlapping
- Background pattern |
| **sina** [78] | | - Arc lines |
| **sohu** [78] | | - Random color for characters
- Random color arc lines
- Character rotation |
| **weibo** [78] | | - Overlapping
- Character rotation
- Color gradient characters |
| **wiki** [78] | | - Styling |

## A.3 Security Features in the Detected CAPTCHAs

**Table 5: CAPTCHA security features. Different features found for text CAPTCHAs in the wild.**

| Features | Usage Rate | Sample CAPTCHAs |
|----------|-----------|-----------------|
| Simple Colors | 22.02% | |
| Background Noise | 56.93% | |
| Excessive Noise | 2.32% | |
| Double Text | 1.25% | |
| Styling | 12.84% | |
| Non-English | 1.62% | |
| Math | 2.97% | |

## A.4 Distribution of CAPTCHA-Enabled Websites

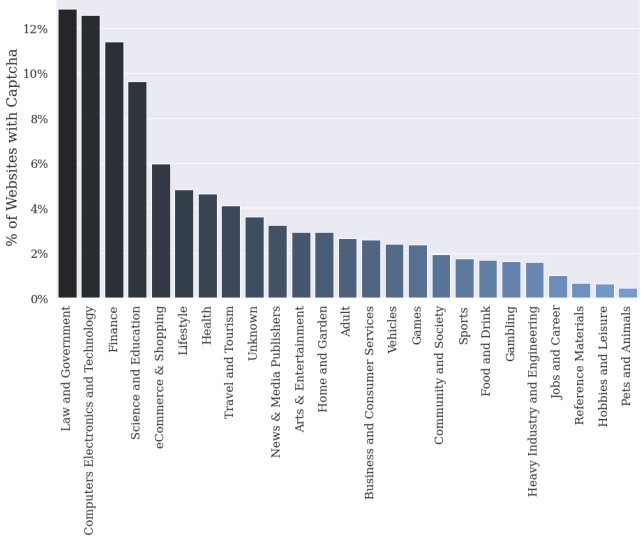

**Figure 5: Distribution of categories for websites that use text CAPTCHAs. Critical categories such as Finance and Health are among the top 10, led by Law & Government.**

## A.5 Uncracked Schemes with our model

## A.6 Evasion Methods in Automated Scanners

## A.7 Notification

Given the number of users and the importance of services offered in the majority of those web applications, we began notifying the

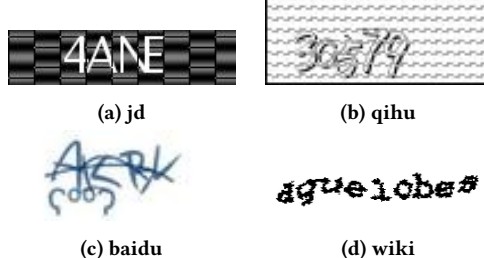

| (a) jd | (b) qihu |
|--------|----------|
| (c) baidu | (d) wiki |

**Figure 6: Schemes samples that our Model failed to crack. Heavy background noise and character overlapping are the most common causes. It can also fail if the captcha presents characters with a different style regarding how the model was trained.**

**Table 6: Evasion Methods in Automated Scanners. The scanners used in the case studies can utilize the following evasive methods using a full-fledged browser on physical machines.**

| Evasion Category | Target Property | Reference | Credential Stuffing | Log4j Exploit |
|------------------|-----------------|-----------|:---:|:---:|
| Browser | User Agent | [19, 20, 50] | ✓ | ✓ |
| | HTTP Headers | [19, 20, 70] | ✓ | ✓ |
| | Plugins list | [19, 20, 49] | ✓ | ✓ |
| | Extensions list | [64, 65, 73] | ✓ | ✓ |
| | Cookies | [7, 70] | ✓ | ✓ |
| | HTML5 Canvas | [2, 19, 41] | ✓ | ✓ |
| | WebRTC | [44] | ✓ | ✓ |
| | Ad Blocker Usage | [43, 44] | ✓ | ✓ |
| Operating System | Timezone | [44, 70] | ✓ | ✓ |
| | Screen Resolution | [19, 20, 49] | ✓ | ✓ |
| | Color Depth | [19, 20] | ✓ | ✓ |
| | List of fonts | [19, 49] | ✓ | ✓ |
| | Platform | [44, 70] | ✓ | ✓ |
| Hardware | WebGL Vendor | [44] | ✓ | ✓ |
| | WebGL Renderer | [44] | ✓ | ✓ |
| Network | IP Cloaking | [27, 32] | ✓ | ✓ |
| | Network Proxies | [32] | ✓ | ✓ |
| Application | Event Listener Fuzzing | [39, 72] | ✓ | ✓ |
| | Pre-trained Solvers | This work | ✓ | ✓ |

system administrators of websites. Since many web applications were using the WHOIS privacy services, we used other services such as contact details scraper from Apify [16] together with manual search for contact information of the websites. After collecting the list of emails, we sent emails to the website owners including a screenshot of the form which employed the mentioned CAPTCHA in two different occasions with minor changes in the text of the emails. Appendix A.8 shows the template of the email we sent to the web application operators.

We received 22% automated or human-generated responses which indicated that the message was received or a ticket was issued based on the email. We received emails to provide more information of the incidents and the examples of possible evasions. At the time of submission, we realized that all the vulnerable pages were removed (e.g., CAPTCHA was removed, the page with the CAPTCHA was

modified or not accessible any more, the inspect elements became disabled).

### A.8 Notification Email

Greetings,

We hope this e-mail finds you well. We are a group of security researchers at *Research Institution*. We have been working on offensive web scanning and their malicious activities on critical web applications.

Our recent analysis shows that the website *example.com* currently uses a vulnerable text CAPTCHA, which is meant to protect against automated bots. This CAPTCHA can be solved automatically, allowing the malicious code to bypass this security measure and gain access to the website's features. This can potentially raise serious security concerns, including the risk of data breaches, website defacements, or other possible abuses (e.g., sending targeted spam and spear phishing attacks). Attached is a screenshot of the observed CAPTCHA for reference. Given the significant role of the website and the number of users, this issue can have a potentially high impact.

With this email, we kindly wanted to bring this issue to your attention and encourage you to use more secure options, such as reCAPTCHA by Google. Please let us know if you need more details about the issue. We would be happy to provide more information and discuss it further.

If you have any questions or concerns regarding this matter, please do not hesitate to contact *researcher* at *researcher@email.*

Best regards,

### A.9 Limitations

In this research, our primary focus was on analyzing English text CAPTCHAs. By concentrating on this specific type of CAPTCHA, we were able to delve deep into its characteristics and challenges. However, it is important to acknowledge that there are various other types of CAPTCHAs exist, each with its own unique features and considerations. One aspect that we were unable to explore thoroughly in this research was non-English CAPTCHAs. Different languages present their own set of linguistic, which can significantly impact the design and effectiveness of CAPTCHAs. Moreover, there are other categories of CAPTCHAs such as math-based CAPTCHAs that we did not cover in this research. Investigating the intricacies of math CAPTCHAs would require incorporating mathematical knowledge into the pipeline stages.

Furthermore, popular CAPTCHA services like reCAPTCHA and hCAPTCHA employ different mechanisms and challenges compared to the text-based CAPTCHAs. These services often incorporate a combination of image recognition, audio challenges, and behavioral analysis to verify users. Researching these types of CAPTCHAs would require significant modifications to our existing pipeline stages, as well as the inclusion of specific datasets and algorithms tailored to handle the unique challenges they present.

Another important point to consider in our analysis of CAPTCHAs in the wild is that we currently do not employ any deep crawling mechanisms. We limit our search to the single page associated with the given URL, focusing solely on CAPTCHAs found within that page. However, we believe that by expanding the crawler to examine multiple pages within each website through deep crawls, we would likely encounter a greater number of text CAPTCHAs.

