# OpenReview forum: "The Matter of Captchas: An Analysis of a Brittle Security Feature on the Modern Web"
_ACM.org/TheWebConf/2024/Conference — TheWebConf24_

### Official Review · Reviewer_T2ox · 2023-11-01

**Novelty:** 5
**Technical Quality:** 7

**Review:**

**Strengths:**
- Interesting approach with a sound methodology
- Large-scale analysis with surprising findings (3K sites with potentially solvable CAPTCHAs)
- Comprehensive dataset of over 15K samples from 20 CAPTCHA schemes
- Large-scale notification campaign to affected sites
- Open-source tool

**Weaknesses:**
- Lack of comparative analysis with reCAPTCHA

Thank you for submitting your work to the web conference 2024. This paper investigates the prevalence of solvable text-based CAPTCHAs in real-world scenarios, identifying over 3,000 websites from CrUX and PILU list with potentially solvable CAPTCHAs. It also assesses the effectiveness of integrated solvers in web scanning, demonstrating their ability to successfully solve one out of five challenges in just one attempt.

Beyond the surprising findings, the approach to crack CAPTCHAs from unseen schemes itself is interesting. It uses object detection (deep learning) to develop a pre-trained solving model that can successfully solve 80% of the CAPTCHA challenges from 13 previously unseen schemes with little fine-tuning in almost two attempts. This further highlights the adaptability of ML-based techniques for circumventing text-based CAPTCHAs.

Another strength of this work is creating a comprehensive catalog of CAPTCHA images, encompassing over 15K samples from 20 different CAPTCHA schemes, which can serve as a baseline for assessing the effectiveness of automated scanners in solving CAPTCHAs for future research.

All in all, this is an excellent paper and I enjoyed reading it. The findings are not just impactful but also unexpected due to the large number of affected websites. The paper can raise awareness about the inadequacy of text-based CAPTCHAs and the dangers they pose, especially to web applications that rely on them, as evidenced by the notification campaign conducted by the authors. Also, the combination of the methodology and the accompanying tool has the potential to inspire future research.

Overall, I don't have any major concerns. The only comment I have is that the paper does not compare the effectiveness of text-based CAPTCHAs with alternative security mechanisms, such as reCAPTCHA v3. A comparative analysis would have provided a broader perspective on the strengths and weaknesses of various CAPTCHA-based mechanisms. After reading this paper, I am now wondering how better is the situation for reCAPTCHA?

## Rebuttal Update

Thank you for answering my comment. I don't have any major concerns, and I am pleased to recommend this work for acceptance.

**Questions:**

-

**Ethics Review Description:**

-

**Reviewer Confidence:**

3: The reviewer is confident but not certain that the evaluation is correct

**Scope:**

4: The work is relevant to the Web and to the track, and is of broad interest to the community

---

### Official Review · Reviewer_PqdE · 2023-11-17

**Novelty:** 2
**Technical Quality:** 3

**Review:**

This work is about integrating publicly available solvers into a Captcha solving system. The authors show that 20% of hitherto unseen captchas can be solved using such techniques. The issue with this paper are two-fold. First, there is nothing new that is being proposed here. It is mostly an evaluation of a publicly available Captcha solver on different types of Captchas. The credit, in this case, should go to the writer of the Captcha solver. The mere integration is not adding a great amount of value (in the reviewer's humble opinion). It is not appearing that the whole is greater than the sum of the parts, unless something is being missed out here.

The user study and the extent of the work are good. It appears a very sound and thorough evaluation has been done.

Some suggestions for improvement:

1. Create a meta-captcha solver that works in real time. Predict the captcha solver that will work for a given Captcha and use it.
2. In fact, you can also predict the relative chances of success of each solver. This will save a lot of time.
3. Can the time it takes to solve a Captcha be used to detect if an automated solution technique is being used or not? Will the mouse movements convey something? This is an interesting line of thinking, where you are using human movement (mouse movement rather) as a side channel. There are works in this space. However, something more can always be done.

**Questions:**

1. Were the websites that have defective captchas (as per your study) informed?
2. Is the combination (existing Captcha solver + authors' test scenario) novel? A detailed statement of novelty is required.

**Ethics Review Description:**

It does not look like the website owners were notified about their faulty (low security) Captchas. Authors to confirm.

**Reviewer Confidence:**

3: The reviewer is confident but not certain that the evaluation is correct

**Scope:**

3: The work is somewhat relevant to the Web and to the track, and is of narrow interest to a sub-community

---

### Official Review · Reviewer_bw5Q · 2023-11-21

**Novelty:** 3
**Technical Quality:** 5

**Review:**

This paper studies the prevalence of a deprecated security feature, namely text-based CAPTCHAs, and also explores how well off-the-shelf, publicly available tools perform in solving them. The authors also train an existing object detection model, which surpasses prior work in solving CAPTCHAs of varying schemes. The proposed model, which can be readily imported in web scanners by cost-sensitive adversaries, can solve ~20% of unknown text-based CAPTCHAs in a single attempt.

I think the paper is well-written and was easy to follow. I also appreciate the authors open-sourcing their artifacts and notifying susceptible websites. Nonetheless, I have some questions/concerns that could help clarify some aspects of the paper and also strengthen its presentation and findings.

Firstly, it's not clear what the paper's novelty and contributions are, since it heavily relies on existing datasets and models. Personally, I find that training an existing model and applying it to a specific problem is not an important contribution in itself. Would the authors be willing to elaborate on this?

Moreover, when explaining how the CAPTCHA catalog was built, some decisions do not seem to be adequately justified. For instance, why did the authors select two colors for the background? It is also mentioned that the number and thickness of background lines and dots as noise were carefully tuned; how was this tuning performed? I assume these features were inferred from the different, known CAPTCHA schemes, but, in any case, should be clarified.

Regarding the training of prior work models, the authors mention their shortcomings and that common hardening techniques could highly impact their accuracy, but no further evaluation is provided. Moreover, the DeepCAPTCHA and DW-GAN models were both trained with a different, simpler scheme than that of the proposed ODM model. In more detail, the ODM model was trained with the "Cat 5" scheme, which is quite more complex than the others and incorporates multiple security features. Therefore, while the ODM model indeed seems to be more sophisticated, it's unclear whether its superiority (partially) stems from training it on a more complex, demanding scheme. How would the other models behave if trained on the same scheme? Also, the number of training samples is not mentioned for the prior work models, nor if a _max attempts_ setting was used for those as well. Finally, DeepCAPTCHA, which was trained on the "Python CAPTCHA" scheme, achieves a 98% accuracy on such CAPTCHAs, while ODM (with 5 attempts) has a success rate of 15% for the same scheme. Similarly, the "Rescator 2" success rate (16%) is also quite low, especially when compared to "Rescator 1" (92%). However, from a visual/user perspective, "Rescator 2" seems simpler than "Rescator 1" and also has less security features. Do the authors have any insights into why these issues occur?

When analyzing the real-world results (5.3), the authors state that 3K websites utilized text-based CAPTCHAs, yet they end up with 1.6K CAPTCHAs for their evaluation. It is unclear why this discrepancy occurs and, therefore, how many websites really incorporate such challenges. On a more general note, I think highlighting the relevance of the problem in the modern Web is crucial, especially given that the whole study revolves around a deprecated feature. While the authors provide some insights on those websites' monthly visits, the analysis still lacks a straightforward indication of the problem's extent, e.g., X% of Alexa's/CrUX Top 1M use text-based CAPTCHAs. Considering only the landing page from the collected URLs to avoid burdening websites with unnecessary requests is commendable; however, I believe a slightly deeper crawl would still be ethically acceptable and, at the same time, beneficial for the study. For instance, the authors could visit the landing pages and leverage a strict, keyword-based approach to uncover further links of interest that point to contact forms, signup/login pages etc. and possibly couple that with a short limit on the number of visited pages. Such an approach would strengthen the paper, as it would better demonstrate the prevalence of the problem.

Finally, I believe that attempting to solve the collected CAPTCHAs in a single attempt most likely undermines the model's effectiveness. In more detail, using the Levenshtein distance (LD) between the predicted and true labels gives some pointers into how the system would perform if it made more attempts, but is not as accurate as actually making more attempts. For instance, for LD <= 2, which accounts for 40-50% of attempted challenges across website categories, the authors state that such cases would be possibly solved if more attempts were made. Indeed, this might truly be the case for a lot of these. On the other hand, in certain cases the missed characters might exhibit specific characteristics (e.g., overlapping, styling etc.) which could prevent the system from correctly identifying them even in subsequent attempts. It would be much more accurate and insightful if the authors gathered and attempted to solve different CAPTCHAs from each URL, by reloading the page a limited number of times (e.g., 5 refreshes). This would paint a more clear picture on the model's capabilities in a realistic setting, where an attacker would conveniently make some extra requests to solve a CAPTCHA, while adding only a few additional requests to the collection phase.

- Pros
    + Demonstrates the ability of a cost-sensitive adversary to solve (deprecated) text-based CAPTCHAs with off-the-shelf models and minimal training.
    + Can solve 20% of unknown, real CAPTCHAs in a single attempt.
    + Open-sources model and artifacts.

- Cons
    + Training of and comparison to prior work is unclear.
    + Highlighting the prevalence of the problem could be significantly improved, considering that such CAPTCHAs are widely deprecated.
    + Evaluation in real world CAPTCHAs could be more thorough by incorporating multiple cracking attempts, better illustrating the proposed model's solving capabilities in real-world settings.

**Questions:**

- Why were DeepCAPTCHA and DW-GAN models trained on simpler schemes? Also, was a _max attempts_ parameter used for their evaluation?

- Why are there 1.6K real-world CAPTCHAs, while the authors mention that 3K websites use them?

- How many websites truly use text-based CAPTCHAs? Can the authors provide further insights, e.g., by correlating with Alexa's or CrUX top 1M websites?

**Reviewer Confidence:**

3: The reviewer is confident but not certain that the evaluation is correct

**Scope:**

3: The work is somewhat relevant to the Web and to the track, and is of narrow interest to a sub-community

---

### Official Review · Reviewer_HAYU · 2023-11-23

**Novelty:** 4
**Technical Quality:** 5

**Review:**

This paper explores the known problem of obsolete (i.e., made of words) CAPTCHAs by claiming that a web scanner embedded with pre-trained solvers can bypass over 20% of websites protected with text-based CAPTCHAs. The authors made use of existing CAPTCHA solvers to conduct a real-world study on 3,000 websites still using old CAPTCHA systems as well as on a catalog of 15,000 CAPTCHA challenges they generated (and made public).

The paper is well written and articulated. The authors meticulously present their work, starting from introducing their goal (i.e., showing that automated scanners can nowadays be equipped with pre-trained solvers to automatically bypass text-based CAPTCHAs) to presenting the tools available in this research space. The paper continues with the experiments that the authors conducted, and concludes with describing *in a correct form* their measurements. While the work is well conducted and goes to the point (the measurement study supports, to a certain extend, the authors' initial hypothesis), I have few comments that I would like to share with the authors:

- While Table 1 reports a consistent amount of previous tools (14 in total), the models were available only for 6 of them. As the authors mentioned in Section 3.2.1, this represented an important limitation. Wouldn't then make more sense to limit the adoption to "fully available" tools rather than incorporating code that resembled more to academic PoCs rather than fully functional tools?
- The solvers seemed to suffer when CAPTCHAs included background noise or overlapping characters as mentioned in Section 4.1 and in Table 2. Is this a problem of the training sets and how can a modern scanner cope with that in real-world testing situations?
- I found the dataset quiet limited in scope. Even though the authors started from a large corpus of pages (i.e., [24] and [66] as mentioned in Section 5.2). they ended up with a dataset of 3,000 websites only. At this point, I am wondering how this study is representative of a general problem of the modern Web. Can the authors do better here to build a more extensive dataset?

**Questions:**

Ref questions above. While the work is interesting, I found some concerns that could be addressed.

**Ethics Review Description:**

Ethics is well addressed.

**Ethics Review Flag:**

Yes

**Reviewer Confidence:**

3: The reviewer is confident but not certain that the evaluation is correct

**Scope:**

4: The work is relevant to the Web and to the track, and is of broad interest to the community

---

### Official Review · Reviewer_hetn · 2023-11-24

**Novelty:** 4
**Technical Quality:** 6

**Review:**

Pros:

- Well-executed measurement of an evadable deprecated security mechanism that is still used on real-world Websites.

- I also very much appreciate that the authors open-sourced an artifact for this study, such that the reader can check how certain parts are implemented and that future work can benefit from the implementation, thanks.

- It is good that the authors took extra care in conducting the measurement ethically, especially I appreciate that the authors contacted the impacted websites about the issue of using an old and bypassable feature.


Cons:

- My biggest concern here is the novelty or the selling point of the approach. I understood that the goal of the paper was not showing the bypassebility of text-based captures, but measuring how easy it is to do this in automated scans. However, this makes this paper more of a measurement than a security paper, especially because of the nice measurement regarding the usage of text-based captures in the wild. Thus, I would recommend moving this to one of the measurement tracks, e.g. search track or submit it to conferences that target Web measurements e.g. IMC.

- In section 5.3 it is mentioned that over 30% of the requested domains were unresponsive domains. Here it would be nice to have more details as unresponsiveness can have multiple reasons. A short explanation of the errors that occurred might also help other works conducting similar scans to deal with this issue.


- Minor Things:

  - In the related Work part about deprecated Web features, the "formal analysis of inconsistent Click-Jacking protection on the web" due to deprecated features like XFO from Calzalvara et al. might also be worth mentioning.

  - In some parts, there are problems with the textual representation, e.g lines that are longer  than the text width (page 3)

**Questions:**

- Are there any responses from web developers that you notified on why they still use this feature? It would be interesting to know here because maybe some laws or rules are requiring some websites to use text-based captures.

- Can you please give us more details about the 30% unresponsive domains that you faced during the measurement crawl?

**Ethics Review Description:**

They carefully conducted their measurements, all good.

**Reviewer Confidence:**

3: The reviewer is confident but not certain that the evaluation is correct

**Scope:**

3: The work is somewhat relevant to the Web and to the track, and is of narrow interest to a sub-community

---

### Decision · Program_Chairs · 2024-01-22

**Decision:**

Accept

**Comment:**

# Summary
 This paper attempts to measure the usage of text-based Captchas, which are used by web applications to protect against automated usage. The paper shows that, by leveraging pre-existing solvers, an automated scanner can solve ~20% of text-based Captchas, thus highlighting the uselessness of this defense.


 # Strengths

 + Interesting problem area: looking at websites that use easily breakable Captchas.
 + Interesting measurement results on usage of breakable Captchas in real-world web sites.
 + Open-source artifact.
 + Careful and ethical study design.

 # Weaknesses

 - Novelty concerns (the paper does not propose Captcha-breaking systems itself).

 # Recommendation

 Overall, the reviewers were mostly unanimous in their appreciate of the results of this paper, and I believe that it will generate interesting discussion at TheWebConf. While there were concerns with novelty, this is outweighed by the strengths of the paper.

 ---